# The Cost-Effectiveness of Homecare Services for Adults and Older Adults: A Systematic Review

**DOI:** 10.3390/ijerph20043373

**Published:** 2023-02-15

**Authors:** Cintia Curioni, Ana Carolina Silva, Jorginete Damião, Andrea Castro, Miguel Huang, Taianah Barroso, Daniel Araujo, Renata Guerra

**Affiliations:** 1Institute of Nutrition, State University of Rio de Janeiro, R. São Francisco Xavier, 524-12º Andar-Bloco E-Sala 12008-Maracanã, Rio de Janeiro 20550-170, Brazil; 2Department of Family Medicine, State University of Rio de Janeiro, Boulevard 28 de Setembro, 77-Vila Isabel, Rio de Janeiro 20551-030, Brazil; 3Hospital Estadual Ary Parreiras, R. Dr. Luiz Palmier, 762-Barreto, Niterói 24110-310, Brazil; 4Health Technology Assessment Unit, Brazilian National Institute of Cancer, R. Marques de Pombal, 125-7º andar-Centro, Rio de Janeiro 20230-240, Brazil

**Keywords:** homecare, hospital care, adults, older adults, economic evaluation

## Abstract

This study provides an overview of the literature on the cost-effectiveness of homecare services compared to in-hospital care for adults and older adults. A systematic review was performed using Medline, Embase, Scopus, Web of Science, CINAHL and CENTRAL databases from inception to April 2022. The inclusion criteria were as follows: (i) (older) adults; (ii) homecare as an intervention; (iii) hospital care as a comparison; (iv) a full economic evaluation examining both costs and consequences; and (v) economic evaluations arising from randomized controlled trials (RCTs). Two independent reviewers selected the studies, extracted data and assessed study quality. Of the 14 studies identified, homecare, when compared to hospital care, was cost-saving in seven studies, cost-effective in two and more effective in one. The evidence suggests that homecare interventions are likely to be cost-saving and as effective as hospital. However, the included studies differ regarding the methods used, the types of costs and the patient populations of interest. In addition, methodological limitations were identified in some studies. Definitive conclusions are limited and highlight the need for better standardization of economic evaluations in this area. Further economic evaluations arising from well-designed RCTs would allow healthcare decision-makers to feel more confident in considering homecare interventions.

## 1. Introduction

The global population is living longer. According to the World Health Organization (WHO), by 2050, people ≥ 60 years are expected to amount to 2.1 billion [1]. However, many of these additional years are not spent in good health or free from disability. Consequently, health systems face increased expenses owing to greater demand [2]. This has sparked interest in ongoing care in the home environment.

Although population aging is a relevant factor that drives the concerns of health systems regarding new models of care, it is not the only one [3]. The demand for care models in the home environment is followed by other equally relevant and eligible health needs, such as care provided to premature babies, children with chronic illnesses and adults with multiple, chronic and degenerative diseases. Thus, the relevance of homecare (HC) services stands out in the current and future health agendas of all healthcare systems, aiming to contribute to the transformation of practices and the configuration of substitutive health networks [4,5].

The increase in HC in several countries follows the interest of those who run health systems in the de-hospitalization process, rationalization of the use of hospital beds, cost reduction and the organization of patient-centered care. Furthermore, this demand poses another challenge for health systems, contributing to a change in the focus on care and the environment provided, aiming for healthcare based on humanization [6,7].

Most people who require long-term care prefer to receive it at home [8]. As healthcare costs rise worldwide, HC represents an opportunity to reduce avoidable adverse events and costs. Furthermore, HC may also offer one means of reducing admission in the hospital demand that result in facilitating the more efficient use of inpatient beds [9].

Systematic reviews have previously been undertaken; however, these have not focused on the comparison between HC services vs. hospital care. In addition, none included only economic evaluations arising from randomized controlled trials. One systematic review indicated that HC was cost-saving when compared to other healthcare modes. However, this review included mainly comparative design studies [10]. Tappenden et al. evaluated the clinical effectiveness of home-based, nurse-led health promotion programs in the United Kingdom. Although they showed that these interventions were clinically effective, only three of the 11 included studies performed cost analyses [11]. Flemming et al. compared the cost-effectiveness of new or improved HC services with usual care and found across the six areas of focus that positive or cost-effective results were reported in two groups (alternative nursing care and reablement/restorative care) [12]. 

Cost-effectiveness studies are considered the gold standard for developing accurate estimates of the value of health interventions to inform decision making [13]. Thus, a broader evaluation of HC services with a critical judgment of the results will help in decision-making regarding the applicability of such services in a unified health system. Therefore, this study aimed to provide an overview of the cost-effectiveness of HC services compared with in-hospital care for adults and older adults.

## 2. Materials and Methods

### 2.1. Registration and Protocol

A protocol for this systematic review was registered on PROSPERO: International prospective register of systematic reviews website (Registration number: CRD42022308742) and the findings were reported according to the Preferred Reporting Items for Systematic Reviews and Meta-Analyses (PRISMA) guidelines [14] (see Appendix A for the PRISMA checklist). 

### 2.2. Review Question

The guiding question of this systematic review was: Are homecare services offered to adults and older adults more cost-effective than hospital services?

### 2.3. Elegibility Criteria

The following inclusion criteria, based on the acronym PICOS, where the acronym represents population (P), intervention (I), comparison (C), outcomes (O) and, (S) study design were used to select studies: (P) studies performed with adults and older adults; (I) studies that addressed HC services (any form of home health care for any disease prevention and treatment, rehabilitation and palliation); (C) the comparison should be hospital care; (O) full economic evaluation examining both the costs and consequences (cost-minimization, cost-effectiveness and cost-utility analyses). Secondary outcomes included: mortality, hospitalizations, readmissions, symptom control, quality of life (QoL), satisfaction with care and costs in a disaggregated way (use of resources with their respective costs); (S) economic evaluations arising from randomized controlled trials (RCTs). 

We excluded studies if the interventions targeted caregivers, including aspects of HC provided outside the home, such as in an outpatient hospital or clinic. In addition, we excluded studies in which the comparison was hospital day care. Studies presented only as abstracts with no subsequent full report of the results were also excluded.

### 2.4. Search Methods for the Identification of Studies

We comprehensively searched electronic databases for records of economic evaluations arising from RCTs of HC interventions compared with hospital care. We performed searches using the MEDLINE (Ovid), Embase (Ovid), Scopus, Web of Science, Cumulative Index to Nursing and Allied Health Literature (CINAHL) and Cochrane Central Register of Controlled Trials (CENTRAL or Cochrane Library) databases from inception to 13 April 2022. The search strategy used was based on the PICO(S) scheme and used in combination with Boolean operators. Appendix A presents all of the used search terms in their combinations. We also searched the reference lists of studies that met the inclusion criteria and reviews to identify additional relevant studies. No restrictions were applied in terms of languages or dates. 

### 2.5. Study Selection

Duplicates were identified using the Endnote X9 Program. All duplicates were removed before the study selection process. Thereafter, the results were transferred to Rayyan QCRI, a systematic review web app. First, a pair of researchers (C.C., A.C.S., M.H. and T.B.) independently screened the titles and abstracts of the found records. The full texts of potentially eligible records were retrieved and independently screened (C.C., M.H. and A.C.S) to confirm inclusion. Disagreements were resolved through discussion by all researchers.

### 2.6. Data Extraction and Risk of Bias Assessment

Pairs of authors (C.C., A.C.F., M.H., T.B., J.D., D.C. and A.C.) independently extracted data and assessed study quality using standardized, piloted data extraction forms in Covidence (a web-based systematic review software program). 

For each trial, the following data were extracted: trial information (author, year of publication, country); type of economic evaluation; funding and conflict(s) of interest; population baseline characteristics (age, sex, etc.); details regarding interventions and comparators; time horizon (the period over which the costs and effects are measured); the economic method used; perspective; year of costs; results/outcomes; and sensitivity analysis results. When information regarding any of the above was unclear or incomplete, we attempted to contact the authors of the original reports to request further details by email.

Regarding the economic evaluation, the quality appraisal of the studies was performed using the Consensus Health Economic Criteria (CHEC) list [15] that focuses only on the methodological quality of economic evaluations. The CHEC list was designed and is recommended for systematic reviews of trial-based economic evaluations. The tool consists of 19 yes-or-no questions for each category. For each question, “yes” was chosen if the study paid sufficient attention to a certain aspect and “no” if insufficient information was available in the article or in other published materials. Positive responses were scored as 1, whereas negative responses were scored as 0. The score for each item was summed and the total CHEC score was transformed to a percentage ranging from 0–100%. A critical appraisal plot (CHEC-list) was produced in Excel 2013.

Disagreements regarding data extraction and critical appraisal were resolved through discussion with all reviewers.

### 2.7. Data Synthesis 

A PRISMA flowchart was used to synthesize the study selection process [14]. Since important differences regarding participants, interventions, diseases and follow-up period were found, a narrative synthesis was used to provide a descriptive summary of the participants’ characteristics and the findings from the included studies. 

## 3. Results

### 3.1. Results of the Search

This literature search retrieved 2969 studies, of which 645 were removed as duplicates, leaving 2324 for title and abstract screening. Fifty-three potentially relevant references were obtained as the full text. At the full-text stage, 39 studies were excluded: 26 assessed only costs and not consequences, 10 did not fulfill the criteria for interventions or comparisons and three studies had the wrong study design. Finally, this review included 14 studies from which data were extracted. No studies were added from the reference lists of studies that met the inclusion criteria. A flow diagram of this process, according to PRISMA guidelines, is presented in Figure 1.

### 3.2. Study Characteristics

Out the 14 studies, eight were conducted in the UK [16,17,18,19,20,21,22,23], three in the Netherlands [24,25,26], one from Sweden [27], one from Italy [28] and one from Iran [29]. The studies were mainly published before 2010. On grouping studies according to the International Classification of Diseases, we found that the most commonly studied illnesses corresponded to the circulatory system (n = 4) [16,17,27,28] and the respiratory system (n = 3) [19,24,25]. The number of participants in the trials varied from 31 to 1055. Six studies randomized older adult patients, two adults and the remainder included both young, middle-aged and older adults. The intervention and comparison covered different treatments related to the care required for the respective disease. 

Five studies included home visits by specialist team [17,20,21,22,23,28,29], three studies included home visits by nurses (nursing care) [18,24,27] and three studies included visit by other specialists [16,19,26]. Additional intervention components included telemonitoring and nursing care [25].

Most of the included studies (n = 11) [16,17,18,19,20,21,22,23,24,25,26] received financial support from public healthcare or non-profit organizations. Two studies did not report funding sources [27,29] and one declared “no funding sources” [28]. 

The characteristics of the included studies are shown in Table 1.

### 3.3. Economic Evaluation

Eight studies performed a cost-effectiveness analysis [17,18,19,23,24,25,27,29] and six performed cost-minimization analyses [16,20,21,22,26,28]. The cost components were organized into three categories: direct medical, indirect and non-medical costs. Effectiveness was assessed in different ways and a range of outcome measures was used in the studies. The most commonly utilized was quality-adjusted life years (QALY), which was present in eight studies [17,18,19,23,24,25,27,29].

Eleven studies presented the cost perspective taken for the evaluation: only societal (n = 4) [16,17,25,29], only the public healthcare payer perspective (n = 2) [19,27] and five studies contained a combination of two perspectives [20,21,22,23,24]. Three studies did not report taking any perspective [18,26,28]. All studies had a time-horizon perspective and the period varied from days to 1.5 years. The most common time periods were three and six months, with four studies each. 

Among the 14 studies included, HC, when compared with hospital care, was cost-saving in seven studies, cost-effective in two and more effective in one. For the management of exacerbations in COPD patients, there was no statistically significant difference between usual home and hospital care strategies [24]. However, exercise programs were cost-effective compared to usual care [19]. When comparing exercise programs, a hospital program was cost-saving (£796 per patient) despite the home-based program incremental effectiveness (0.04 QALY/patient) [19].

Four studies focused on diseases of the circulatory system, two assessed acute conditions (stroke and heart infarction) and two assessed chronic conditions (disabling stroke and heart failure) [17,18,27,28]. In three of them, HC strategies were cost-saving [17,27,28], even though Kalra’s results showed that stroke units (hospitals) were more effective (0.076 QALY/patient) [17].

For conditions that required acute care, including multiple events, the effectiveness of HC and hospital programs was similar [20,21,22,23]. However, hospital costs were higher, resulting in cost-saving HC strategies. The incremental costs of hospital programs were £205.58, £776 and £2840.00 per patient, respectively [20,21,23]. HC strategies for patients suffering from diabetic foot ulcers are cost-effective compared to hospital care. The incremental cost-effectiveness ratio (ICER) was US $117,300 per QALY [29].

Despite the effectiveness of HC and hospital treatment for patients requiring long-term injectable agents for the treatment of tuberculosis being similar, the hospital-at-home scheme was less costly than receiving care in the hospital—a cost difference of US $602.3 [16]. The same was observed in domiciliary antenatal fetal monitoring for high-risk pregnancies. Domiciliary monitoring is effective and reduces costs by one half [26].

Eight studies performed a sensitivity analysis [17,19,20,22,23,24,27,29]. In four of them, the results were not altered [19,20,23,29]. Goosens’s study [24] found that, from a societal perspective, the cost rose due to HC disappearing almost entirely. From a healthcare standpoint, the finding that HC led to cost savings was surrounded by almost no uncertainty. In Kalra’s study [17], if decision-makers were not willing to raise costs for QALY gains, there would be a 59% probability that HC would be the most cost-effective (i.e., optimal). This probability fell with increasing levels of willingness to pay for QALY gains, but remained higher than the other two strategies. In the work of Patel [22], sensitivity analysis altered the obtained values, but HC remained cost saving. Finally, reducing the length of hospital-at-home care changed the difference in total healthcare costs for patients with chronic obstructive airway disease.

All economic evaluations are described in Table 2.

### 3.4. Secondary Outcomes

All studies reported mortality as an outcome, but none detected differences between HC and the hospital. The same was observed for the seven studies that evaluated hospital readmissions [16,17,18,19,22,24,26]. Nine studies [17,18,19,20,21,22,23,24,25] applied a scale to evaluate quality of life (QoL). In one study, patients in the hospital-at-home group reported a significantly greater improvement in QoL compared to those in the hospital group [22]. 

Only two studies reported satisfaction as an outcome [17,20]. In one study, hospital-at-home patients perceived higher levels of involvement in decision-making [20]. In the other study, a significant difference favoring homecare was observed for being able to talk about problems with professionals, information on the nature and cause of stroke, the organization of care, support and the amount of contact with the specialist [17]. Adverse events were described in only four studies [16,19,28,29], three of them showing a higher number of adverse events in the hospital group [16,28,29]. Details about the secondary outcomes are listed in Table 3.

### 3.5. Quality Appraisal of the Included Studies

Figure 2 summarizes the appraisal of reporting quality for each study using the CHEC list. Overall, there were some limitations to the quality of the identified studies, particularly concerning the poor consideration of the methods of outcome valuation (Q9), the discount on future costs and outcomes (Q14), the lack of incremental analysis and sensitivity analysis. As highlighted by Figure 2, all the studies fulfilled the items regarding the clear description of study population; the clear description of competing alternatives; the identification of all important and relevant outcomes; and the appropriated outcomes measures. Only two studies fulfilled all the assessed criteria [17,23]. Three studies fulfilled less than 70% of the assessed items [21,26,28]. See Appendix A for the details of quality appraisal of the included studies. 

Q1. Is the study population clearly described?

Q2. Are competing alternatives clearly described?

Q3. Is a well-defined research question posed in an answerable format?

Q4. Is the economic study design appropriate to the stated objective?

Q5. Is the chosen time horizon appropriate for including the relevant costs and consequences?

Q6. Is the actual perspective chosen appropriate?

Q7. Are all important and relevant costs for each alternative identified?

Q8. Are all costs measured appropriately in physical units?

Q9. Are costs valued appropriately?

Q10. Are all important and relevant outcomes identified?

Q11. Are all outcomes measured appropriately?

Q12. Are outcomes valued appropriately?

Q13. Is an incremental analysis of costs and outcomes performed?

Q14. Are all future costs and outcomes discounted appropriately?

Q15. Are all important variables, whose values are uncertain, appropriately subjected to sensitivity analysis?

Q16. Do the conclusions follow from the data reported?

Q17. Does the study discuss the generalizability of the results to other settings and patient or client groups?

Q18. Does the article indicate that there is no potential conflict of interest with the study researcher(s) and/or funder(s)?

Q19. Are ethical and distributional issues discussed appropriately?

## 4. Discussion

Very few studies have considered the costs and outcomes of home healthcare interventions compared with hospital care for disease prevention, treatment, rehabilitation and palliation. The evidence suggests that home healthcare interventions are likely to be cost saving and as effective as hospital care interventions. However, the studies included in this review differ in terms of the methods used, types of costs and patient populations of interest. Thus, it was difficult to directly compare the individual results. 

Although one might expect that, for acute emergency conditions or those related to surgical processes, the hospital would be the most cost-effective intervention, this review revealed that the studies showed similar effectiveness, except for Kalra’s study [17], where one of the options (hospitals) was more effective in treating stroke patients. In addition, in general, HC was cost-saving, except in the work of Shepperd [22]. This study was carried out in the 1990s in the UK, which was going through an important health system reform at the time, introducing the idea of care centered on individual needs [30]. This could be a possible explanation for the high cost of HC intervention. Furthermore, new technologies can lower the costs of HC.

Regarding effectiveness, results from other reviews found similar results and also report heterogeneity and scarcity of methodologically adequate studies. In a systematic review, Leong et al. showed that HC generally leads to similar or improved clinical outcomes compared to inpatient treatment [31]. For patients with decompensated heart failure, HC appears to increase the time to readmission and improve QoL compared with routine hospitalization. However, HC did not significantly reduce readmission or mortality [32]. In a systematic review of patients with chronic diseases who went to the emergency department, HC lowered the risk of hospital readmission and long-term care admission compared to in-hospital care. The mortality risk was similar between the two groups [33].

In terms of QoL, the findings were still similar. QoL is a broad and complex concept, defined as one’s perception of their position in life, culture and value systems in the context of life, as well as in relation to objectives, expectations, standards and concerns [34]. In this sense, obtaining a high QoL and a high level of HC services is challenging.

Despite the variety of diseases, perspectives, costs and outcomes, most studies have shown results favoring HC modalities. Nevertheless, it is necessary to understand and analyze each respective disease because it will impact demands that could be met at a better cost in the hospital environment. Important outcomes, such as adverse events and satisfaction, were assessed in a few studies. Care could emphasize practical wisdom in a close relationship with techno-scientific knowledge; that is, a set of instrumental actions considered adequate and correct by the actors involved. Hence, this implies considering human subjectivity, understanding the pursuit of happiness and ways of living throughout the course of illness [35].

A societal or health system perspective was adopted in most of the selected studies. Almost half of the study populations comprised older adults living in high-income countries. The societal impact differed between the retired and economically active populations. This should be considered in future research. With few exceptions, most diseases evaluated can have a significant impact on the productivity of the affected individuals or caregivers, both by impeding them from working and affecting mental well-being [36]. Decreased productivity can translate into lost income, which impacts people with illnesses and their families. Only two studies contained both the societal and health system perspectives [23,24]; however, if the inclusion of societal costs led to substantial changes in the outcomes, then this matter was poorly explored.

Notably, most of the included studies were conducted in the UK, which has a universal healthcare system called the National Health Service, as well as from the Netherlands, which has had a hybrid healthcare system (a multi-payer system based on managed competition between private insurers and providers) since 2006 [37,38]. Healthcare expenditure is rising worldwide and continues to be a concern for health systems [39]. There is an urgent need for cost-effectiveness assessments to support policies and actions. No studies have been performed in countries with only private health systems. There is apprehension about private equity firms that now own several of the largest HC chains in several countries [40]. The widespread use of predatory financial practices by these entities has raised concerns because they can prioritize profits over quality of care [41].

Finally, cost-effectiveness analysis was reduced when there was no integration between the levels of healthcare. An integrated healthcare system is essential to enable a connected, holistic view of the patient’s journey across different care settings such as hospitals, outpatient care and homes. 

### 4.1. Strengths and Limitations

The strengths of this review include a registered protocol that addresses the items on the PRISMA checklist. Furthermore, we performed a comprehensive search strategy that was not limited by year or language. Two reviewers independently selected and extracted the studies and assessed their quality.

The studies were heterogeneous and there was considerable variation in their methods, outcomes and patient populations of interest, which made it difficult to compare them. In addition, important outcomes such as QoL, satisfaction and adverse events were not measured in most of the selected studies. Despite every effort being made to identify studies on this topic, the presence of publication bias cannot be excluded.

### 4.2. Future Research

Future studies should explore patient characteristics that impact the cost-effectiveness of home care, such as conditions of patients (acute or chronic conditions), age effect, household, financing model and coverage of national health systems. Further economic evaluations arising from well-designed RCTs with improved reporting would allow healthcare decision-makers to feel more confident in considering home healthcare interventions.

## 5. Conclusions

Current evidence for home healthcare interventions suggests that they are likely to be cost saving and as effective as hospital care interventions. Definitive conclusions are limited by quantity according to different conditions and quality, as this review identified some methodological constraints in the existing literature, highlighting the need for better standardization of economic evaluations in this area. 

## Figures and Tables

**Figure 1 ijerph-20-03373-f001:**
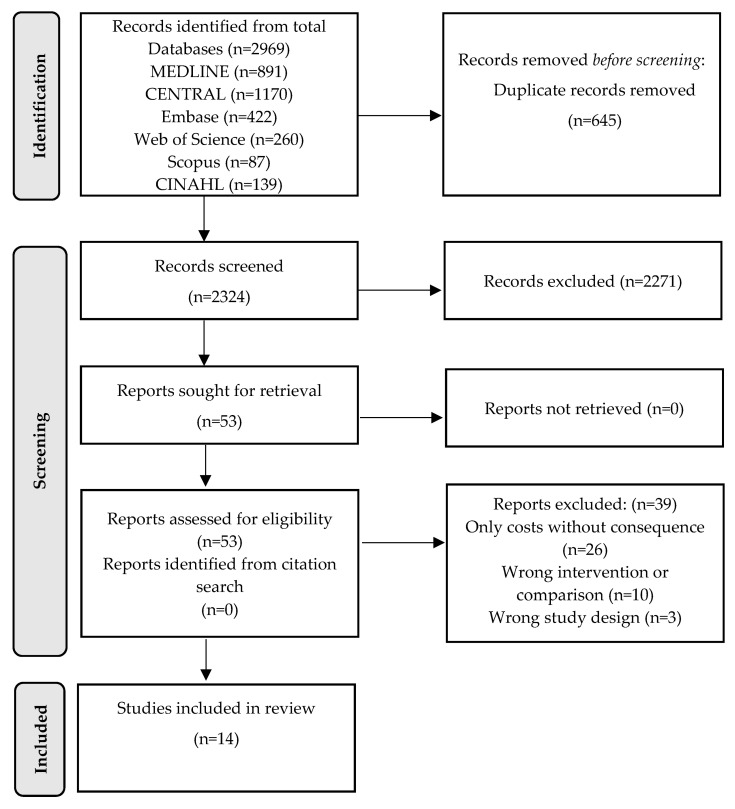
PRISMA study flow diagram for search up to 13 April 2022.

**Figure 2 ijerph-20-03373-f002:**
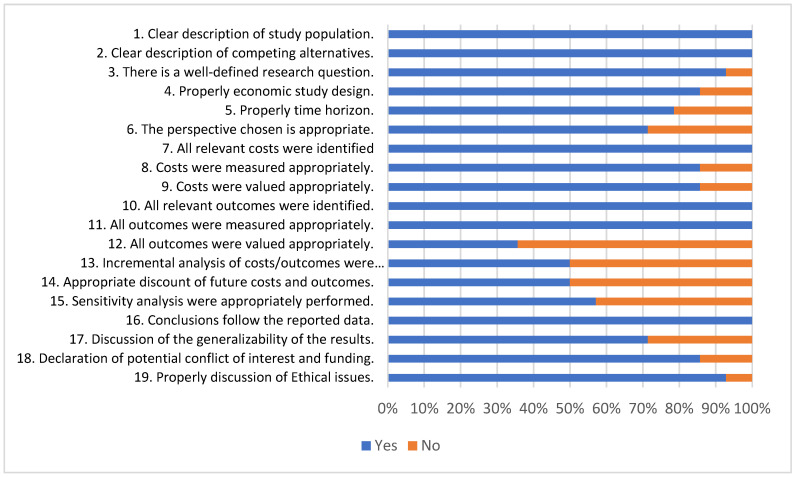
Appraisal of economic evaluations using the quality CHEC list.

**Table 1 ijerph-20-03373-t001:** Characteristics of the included studies.

Study/Country	Disease	Intervention—Homecare	Comparison—Hospital	Sampling	Age Range	Funding
Certain infectious and parasitic diseases						
Cohen et al. [16]/ UK	Recurrent or drug-resistant tuberculosis	Participants received home-based care from guardians trained to deliver intramuscular streptomycin.	Participants were admitted to the hospital for 60 days.	205	30–44	Public/Non-profit
Diseases of the circulatory system						
Kalra et al. [17]/UK	Disabling stroke	Patients were managed in their own homes and care was provided by a specialist team (doctor, nurse and therapists), with support from district nursing and social services for nursing and personal care needs. Patients were under the joint care of the stroke physician and the GP. Each patient had an individualized, integrated care pathway outlining activities and the objectives of treatment, which were reviewed at weekly multidisciplinary meetings. This support was provided for a maximum of 3 months.	The stroke unit provided 24-h care delivered by a specialist multidisciplinary team based on clear guidelines for acute care, the prevention of complications, rehabilitation and secondary prevention. The stroke team involved management on general wards with specialist team support. The team undertook stroke assessments and advised ward-based nursing and therapy staff on acute care, secondary prevention and rehabilitation aspects.	457	67–84	Public/Non-profit
Patel et al. [27]/Sweden	Chronic heart failure	Patients are visited at home daily or on alternate days by the specialist nurse as determined by the patient’s health status. Home visits were terminated when a patient: (1) was symptomatically stable or improving; (2) had stable or falling weight; (3) had no signs of pulmonary rales; and (4) had no edema above the ankle.	The patients were treated in accordance with hospital treatment guidelines.	31	67–87	NA
Ricauda et al. [28]/Italy	First acute ischemic stroke	Patients received a HC program that emphasized a task- and context-oriented approach, which recommends that the patient perform guided, supervised and self-directed activities in a functional and familiar context. The standard daily intervention consisted of one visit by a physician, a nurse and a physical therapist.	The inpatient group received routine hospital rehabilitation services, which allocated physical therapists to patients assigned to both groups of the trial.	120	74–89	None
Taylor et al. [18]/ UK	Uncomplicated acute myocardial infarction	Patients were seen during hospital admission by a cardiac rehabilitation nurse and issued the Heart Manual to use over six consecutive weeks.	Patients attended outpatient classes once a week for 8–10 weeks. Classes lasted 2 h each and were conducted in groups of 8–10 people at the local hospital or, for a small number of patients, in one of the two community centers.	104	51–76	Public/Non-profit
Diseases of the respiratory system						
Cox et al. [19]/UK	Chronic obstructive pulmonary disease	The intervention consisted of eight exercises (adapted to each participant’s capability). Four sessions over two weeks were delivered by a physiotherapist in the patient’s home.	A cycle ergometer was used to deliver exercises at hospital bedside. The prescription (cycle workload) was set by a physiotherapist. The patient completed 16 revolutions of the bike for both sets of limbs, three times a day for 5 consecutive days.	58	55–79	Public/Non-profit
Goossens et al. [24]/The Netherlands	Chronic obstructive pulmonary disease	For the first three days, all patients received usual hospital care. Starting on the fourth day, community nurses visited and provided care at least once or up to three times on the day of discharge and over the following three days. During the four days of home treatment, the emphasis was on recovering from exacerbation of symptoms.	Usual hospital care	139	57–79	Public/Non-profit
van den Biggelaar et al. [25]/The Netherlands	Neuromuscular disease or thoracic cage disorder	Patients received mechanical ventilation at home.	Patients started home mechanical ventilation in the hospital.	96	42–70	Public/Non-profit
Endocrine, nutritional and metabolic diseases						
Jafary et al. [29]/Iran	Diabetic foot ulcers	Treatment was performed according to the clinical guidelines approved by Iran’s Ministry of Health. The home visit team consisted of a GP and 3 nurses. Following the initial home visit, additional home visits were conducted at least once a week. Patients could contact the HC providers when the need arose.	Conventional care at the hospital.	120	48–73	NA
Multiple health conditions						
Coast et al. [20]/UK	Hospitalized but medically stable elderly patients	Patients able to receive early discharge from the hospital were allocated to home-based rehabilitative care provided by a multi-professional team (nurse, physiotherapist, occupational therapist and support workers).	Patients received routine hospital care with discharge at the usual time.	241	72–84	Public/Non-profit
Jones et al. [21]/UK	Mix of medical conditions	A GP maintained medical responsibility for 14 days. Multidisciplinary care (nurses, physiotherapists, occupational therapists, generic healthcare workers and cultural link worker) provided between four and 24 h of care per day. They provided access to equipment needed for home nursing such as hospital beds, mattresses, commodes, etc.	Acute hospital admission.	199	77–89	Public/Non-profit
Shepperd et al. [22]/UK	Mix of medical conditions	Care consisted of observation, the administration of (intravenous) drugs, nursing care (in addition to support from other professionals) 24 h a day in the patient’s home if necessary and the rehabilitation of patients at home.	Inpatient hospital care: patients recovering from a hip replacement, a knee replacement, or a hysterectomy; patients with chronic obstructive airway disease; and elderly patients with a mix of medical conditions.	242	58–76	Public/Non-profit
Singh et al. [23]/UK	Acute inpatient hospital care	The HC was based on an evaluation of CGA services received previously in the hospital and subsequently being provided at home.	An inpatient care group received CGA services.	1055	76–90	Public/Non-profit
Pregnancy, childbirth and the puerperium						
Birnie et al. [26]/ The Netherlands	High-risk pregnancies	A midwife performed a daily visit, conducted a cardiotocography and transmitted the tracings to the hospital. Women were seen weekly at the antenatal clinic.	Women were hospitalized and monitored daily. If necessary, they received additional diagnostics or treatment.	150	24–37	Public/Non-profit

CGA: comprehensive geriatric assessment; COPD: chronic obstructive pulmonary disease; GP: general practitioner; HC: homecare; NA: not applicable.

**Table 2 ijerph-20-03373-t002:** Results of economic evaluation of included studies.

Study	Economic Evaluation Type	Effectiveness Outcomes	Perspective/Time Horizon	Cost Description/Year of Costs	Costs	QALY	Cost-Effectiveness ICER/Cost/QALY	Synthesis
Certain infectious and parasitic diseases								
Cohen et al. [16]	Cost-minimization analysis	Successful treatment (alive and upon receiving treatment)	Societal/ 7 weeks	Direct medical; non-medical (food/diet, transportation, visits); indirect (income lost due to illness). Year: 2014	HC: US $498.0 Hosp: US $1100.3 Difference: US $ −602.3	Not reported	Not reported since effectiveness was similar	HC: cost-saving
Diseases of the circulatory system								
Kalra et al. [17]	Cost-effectiveness analysis	Mortality or institutionalization	Societal/12 months	Direct medical (hospital doctors, nursing care, physiotherapy, occupational therapy, psychologist, dentist, etc., consultations); non-medical (food/diet, visits, social work, companion). Year(s): 1997/1998	Mean costs HC: £6840 Hospital 1: £11,450 Hospital 2: £9527 Difference HC × Hospital 1: £4609.94 Difference HC × Hospital 2: £2686.78 Cost per day alive: HC: £36.07 Hospital 1: £37.98 Hospital 2: £50.90 Difference HC × Hospital 1: £ −1.91 Difference HC × Hospital 2: £ −14.83	HC: 0.221 Hospital: 0.297 Difference: 0.076	ICER for hospital £64,097 Homecare dominant Probability of avoiding death/institutionalization HC: 77.86 Hospital 1: 87.16 Hospital 2: 69.39 Cost per death/institutionalization avoided: HC: 0.46 Hospital 1: 0.44 Hospital 2: 0.73	Hospital: effective HC: cost-saving
Patel et al. [27]	Cost-utility analysis	QALY	Public health system/ 12 months	Direct medical (nursing care, consultations); non-medical (transportation). Year: NA	Home: €1122 Hospital: €5110 Difference: € −3988	Home: 0.71 Hospital: 0.64	Cost/QALY lower in home group, but values are not shown and the difference did not reach statistical significance.	HC: cost-saving
Ricauda et al. [28]	Cost-minimization analysis	Mortality;residual functional impairment;neurological deficits	NA/6 months	Direct medical (hospital doctors, nursing care, physiotherapy, occupational therapy, psychologist, dentist, etc., medication, labs/diagnosis). Year: NA	Per day HC: US $163.0 Hospital: US $275.6 Difference: US $ −112.6	Not reported	Not reported since effectiveness was similar for all outcomes.	HC: cost-saving
Taylor et al. [18]	Cost-effectiveness analysis	QALY	NA/9 months	Direct medical (hospital doctors, nursing care, medication, labs/diagnosis); non-medical (transportation). Year(s): 2002/2003	HC: £3279 Hospital: £3201 Difference: £78	Home: 0.74 Hospital: 0.81 Difference: −0.06	Mean incremental cost per QALY: £ −644	Difference not significant
Diseases of the respiratory system								
Cox et al. [19]	Cost-effectiveness analysis	QALY	Public health system/3 months	Direct medical (physiotherapy, occupational therapy, psychologist, dentist, etc., equipment); non-medical (transportation, visits, social work). Year(s): 2015/2016	HC: £4757 Hospital: £3961 Difference: £796	HC: 0.149 Hospital: 0.145 Difference: 0.04	ICER: HC and hospital dominant Probability more effective (QALYs) Home: −0.62 Hospital: −0.56	HC: more effective Hospital: cost-saving
Goossens et al. [24]	Cost-effectiveness and cost-utility	Incremental change in CCQ score; QALY	Societal Public health system/3 months	Direct medical (hospital doctors, nursing care, medication); non-medical (transportation); indirect (productivity losses); other (readmission). Year: 2009	Health care HC: €4129 Hospital: €4297 Difference: €168 Societal HC: €6304 Hospital: €5395 Difference: €880	HC: 0.170 Hospital: 0.175 Difference: −0.05	Health care perspective ICER: €31,111 Societal perspective ICER: Hospital dominant	Difference not significant from either perspective
van den Biggelaar et al. [25]	Cost-effectiveness analysis	Change in arterial CO_2_; QoL	Societal/6 months	Direct medical (nursing care, physiotherapy, occupational therapy, psychologist, dentist, etc., equipment); non-medical. Year(s): 2017/2018	HC: €1500 Hospital: €4725 Difference: €3225	HC: 0.26 Hospital: 0.25 Difference: 0.01	Not reported	HC: cost-saving
Endocrine, nutritional and metabolic diseases								
Jafary et al. [29]	Cost-effectiveness analysis	QALY	Societal/6 months	Direct medical (hospital doctors, nursing care, medication); non-medical (transportation, visits); indirect (productivity losses). Year: 2017	HC: US $1545 Hospital: US $3891 Difference: US $ −2346	Home: 0.31 Hospital: 0.29 Difference: 0.02	ICER: −117.300	HC: cost-effective
Multiple health conditions								
Coast et al. [20]	Cost-minimization analysis	QoL; satisfaction; physical functioning; length of stay; mortality	Individuals Public health system/3 months	Direct medical (hospital doctors, nursing care, physiotherapy, occupational therapy, psychologist, dentist, etc., medication, labs/diagnosis, equipment); non-medical (food/diet, transportation, visits, social work, companion). Year: 1996	HC: £2516 Hospital: £3292 Difference: £ −776	Not reported	Not reported since effectiveness was similar	HC: cost-saving
Jones et al. [21]	Cost-minimization analysis	Mortality and change in health status (Barthel index, Sickness Impact Profile 68, EuroQol, Philadelphia Geriatric Morale Scale)	Individuals Public health system/8 months	Direct medical (hospital doctors, nursing care, physiotherapy, occupational therapy, psychologist, dentist, etc.); non-medical (transportation, visits, social work). Year: NA	HC: £3671.28 Hospital: £3876.86 Difference: £ −205.58	Not reported	Not reported since effectiveness was similar	HC: cost-saving
Shepperd et al. [22]	Cost-minimization analysis	QoL; mortality; Readmission	Individuals Public health system/3 months	NA Year(s): 1994/1995	Hip replacement HC: £911.39 Hospital: £815.70 Difference: £95.69 Knee replacement HC: £1461.62 Hospital: £1375.36 Difference: £86.26 Hysterectomy HC: £771.78 Hospital: £679.39 Difference: £92.39 Elderly medical HC: £1705.32 Hospital: £1388.76 Difference: £316.56 Chronic obstructive airways disease HC: £2379.67 Hospital: £1247.64 Difference: £1132.03	Not reported	Not reported since effectiveness was similar	Hospital: cost-saving for the studied conditions
Singh et al. [23]	Cost-effectiveness analysis	QALY; mortality; QoL	Societal Public health system/6 months	Direct medical (nursing care, physiotherapy, occupational therapy, psychologist, dentist, etc., medication, equipment); non-medical (transportation, visits); other (loss of income). Year(s): 1994/1995	HC: £19,067 Hospital: £21,907 Difference: £ −2840	Home: 0.245 Hospital: 0.247 Difference: −0.002	Probability of home intervention being cost-effective at threshold of £20,000 per QALY: 97%	HC: cost-effective
Pregnancy, childbirth and the puerperium								
Birnie et al. [26]	Cost-minimization analysis	Pretchtl neurologic optimality score: HC: 58.1 Hospital: 57.7 Mortality	NA/Days until birth	Direct medical (nursing care, medication, labs/diagnosis); non-medical (food/diet, transportation, visits); other (monitoring sessions, professional home help, informal family care, premature pregnancy leave). Year: 1993	HC: US $1521 Hospital: US $3558 Difference: US $2037	Not reported	Not reported since effectiveness was similar for both outcomes	HC: cost-saving

HC: homecare; QALY: quality-adjusted life years; QoL: quality of life.

**Table 3 ijerph-20-03373-t003:** Results of secondary outcomes.

Study	Mortality	Readmissions	QoL	Satisfaction	Adverse Events
Certain infectious and parasitic diseases					
Cohen et al. [16]	HC: 13/83 (15.7%) Hospital: 11/69 (13.9%)	HC: 6/83 Hospital: 2/69	Not reported	Not reported	HC: 34/103 (33.0%) Hospital: 56/101 (55.4%)
Diseases of the circulatory system					
Kalra et al. [17]	HC: 21/144 (14.6%) SU: 13/152 (8.6%) ST: 34/149 (22.8%)	HC: 13/144 (9.0%) SU: 8/152 (5.3%) ST: 11/149 (7.4%)	HC: 75 SU: 80 ST: 80	Significant difference favoring HC for: being able to talk about one’s problems with professionals; information on the nature and cause of stroke; the organization of care; support; and the amount of contact with the specialist.	Not reported
Patel et al. [27]	HC: 2/13 (15%) Hospital: 2/18 (11%)	Not reported	Not reported	Not reported	Not reported
Ricauda et al. [28]	Not reported	Not reported	Not reported	Not reported	HC: 28/60 (46.7%) Hospital: 34/60 (56.6%)
Taylor et al. [18]	HC: 4/60 (6.7%) Hospital: 1/44 (2.3%)	HC: 9/60 (15%) Hospital: 6/44 (14%)	HC: 4.66 Hospital: 4.87	Not reported	Not reported
Diseases of the respiratory system					
Cox et al. [19]	HC: 0/15 Hospital: 0/14	HC: 10/15 (66.7%) Hospital: 9/12 (75%)	HC: 0.6 Hospital: 0.5	Not reported	HC: 15/15 (100%) Hospital: 13/14 (93.0%)
Goossens et al. [24]	HC: 1/70 Hospital: 1/69	HC: 17/70 (25%) Hospital: 17/69 (24%)	HC: 0.677 Hospital: 0.672	Not reported	Not reported
van den Biggelaar et al. [25]	HC: 7/47 (15%) Hospital: 5/49 (10.2%)	Not reported	HC: 50 Hospital: 51.7	Not reported	Not reported
Endocrine, nutritional and metabolic diseases					
Jafary et al. [29]	Within 6 months HC: 6/30 (20%) Hospital: 15/90 (16%)	Not reported	Not reported	Not reported	HC: 0/30 (0.0%) Hospital: 16/90 (17.0%)
Multiple health conditions					
Coast et al. [20]	HC: 12/160 (7.5%) Hospital: 6/81 (7.4%)	Not reported	Only reported the difference: −0.04 (−0.13 to 0.06)	Excellent: HC: 50.7% (79/155) Hospital: 44.6% (31/70)	Not reported
Jones et al. [21]	HC: 26/101 (25.7%) Hospital: 30/96 (31.3%)	Not reported	HC: 0.64 Hospital: 0.63	Not reported	Not reported
Shepperd et al. [22]	Hip replacement: HC: 0/37 (0%) Hospital: 1/49 (2%)	Hip replacement: HC: 2/37 (5%) Hospital: 1/49 (2%)	Hip replacement: HC: 3.91 * Hospital: 3.2 *	Not reported	Not reported
	Knee replacement: HC: 0/47 (0%) Hospital: 0/39 (0%)	Knee replacement: HC: 4/47 (9%) Hospital: 1/39 (3%)	Knee replacement: HC: 3.35 Hospital: 3.25		
	Hysterectomy: HC: 0/114 (0%) Hospital: 0/124 (0%)	Hysterectomy: HC: 7/114 (6%) Hospital: 13/124 (10%)	Hysterectomy: HC: 3.18 Hospital: 3.34		
	Elderly medical: HC: 9/50 (18%) Hospital: 4/46 (9%)	Elderly medical: HC: 13/50 (26%) Hospital: 5/46 (11%)	Elderly medical: HC: 2.97 Hospital: 3.23		
	Chronic obstructive airway disease: HC: 3/15 (20%) Hospital: 3/17 (18%)	Chronic obstructive airway disease: HC: 8/15 (53%) Hospital: 6/17 (35%)	Chronic obstructive airway disease: HC: 3.54 Hospital: 2.82		
Singh et al. [23]	HC: 114/673 (15%) Hospital: 58/326 (15%)	Not reported	HC: 0.4334 Hospital: 0.4337	Not reported	Not reported
Pregnancy, childbirth and the puerperium					
Birnie et al. [26]	HC: 1/76 Hospital: 1/74	HC: 47/76 (61.8%) Hospital: 69/74 (93.2%)	Not reported	Not reported	Not reported

HC: homecare; QoL: quality of life.

## Data Availability

No new data were created or analyzed in this study. Data sharing is not applicable to this article.

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
