# Peer review of "The Cost-Effectiveness of Homecare Services for Adults and Older Adults: A Systematic Review"

_ijerph, 2023, doi:10.3390/ijerph20043373_

Round 1

Reviewer 1 Report

This study is well-conducted and well-organized. Three minor suggestions for the authors’ reference.

1.      2.7 Data synthesis section: The authors stated that ‘…Since the heterogeneity of the information collected,…’ How this assertion was made? More convincing evidence may be required.

2.      Based on the title of this study, the authors intended to analyze the cost-effectiveness of adults and older adults, but ‘age’ is not listed as a criterion for selecting studies. Further, the authors may consider if it’s necessary to compare the cost-effectiveness of homecare services between adults and older ones since they may be different in the consumption of healthcare resources.

3.      Typos: 3.3. Economic evaluation section: Second paragraph, lines 3 – 4, (‘did not’) is duplicated and should be corrected.

Author Response

Point 1. This study is well-conducted and well-organized. Three minor suggestions for the authors’ reference.

Response 1: We appreciate it.

Point 2. 2.7. Data synthesis section: The authors stated that ‘…Since the heterogeneity of the information collected,…’ How this assertion was made? More convincing evidence may be required.

Response 2: Thanks for your comment. Heterogeneity refers to any kind of variation across studies, for example, differences associated with the participants, interventions, or outcomes. Furthermore, heterogeneity might involve considering whether very different populations are receiving the intervention across studies, if the interventions are different in important ways across studies, or whether the control or other comparison groups are different across the included studies.

In this sense, participants, interventions, comparisons, and outcomes need to be taken into consideration to determine whether they are similar enough to ensure a clinically meaningful answer when the results are grouped. If studies are very dissimilar on some or all of these factors, it may be preferable not to pool the results.

In our review, the participants, the diseases evaluated, the interventions, and the follow-up period were very different. For this reason, the results were only presented descriptively and narratively. We changed the following sentence to make clear this point (page 4, lines 146-8):

“Since important differences regarding participants, interventions, diseases, and follow-up period were found, a narrative synthesis was used to provide a descriptive summary of the participants’ characteristics, and the findings from the included studies.

Point 3. Based on the title of this study, the authors intended to analyze the cost-effectiveness of adults and older adults, but ‘age’ is not listed as a criterion for selecting studies. Further, the authors may consider if it’s necessary to compare the cost-effectiveness of homecare services between adults and older ones since they may be different in the consumption of healthcare resources.

Response 3: We included only studies with adults and older adults, as described on page 2, line 87: “(P) studies performed with adults and older adults”.

As expected increasing aging could demand more homecare services. This could be explained by the higher frailty and chronic health conditions. However, the costs associated with the homecare will depend, especially, on the demands required by the treatment of the specific disease. None of the included studies with adults and older adults assessed costs stratifying by age. For this reason, unfortunately, we could not explore in this review if age could interfere with the consumption of healthcare resources at home.

As a suggestion of the other Reviewer, we add the need to explore the age effect on the costs in future studies (lines 359-365)

Point 4. Typos: 3.3. Economic evaluation section: Second paragraph, lines 3 – 4, (‘did not’) is duplicated and should be corrected.

Response 4: Thanks for that. Now it is corrected in the text.

Reviewer 2 Report

Thanks to the authors for their submission to IJERPH. The aim of this manuscript is to conduct a systematic review to provide an overview of the cost-effectiveness of health services compared to inpatient care for adults and older people. I fully acknowledge the time and effort invested in searching and screening the data, analysing and synthesising the results, and subsequently writing the manuscript. However, I believe that the authors need to address and clarify some points in the manuscript.

Kind regards,

COMMENTS

INTRODUCTION

I congratulate the authors on this section. It is well written, concise and includes all the information necessary to contextualise using up-to-date and relevant literature and justify the need for the review.

METHODS

The authors declare that they follow the PRISMA guidelines in this systematic review; however, any reference to these guidelines is included in the manuscript. Please insert the appropriate reference when referring to PRISMA.

Please rename the subsection "Criteria for considering studies included in this review" to "Eligibility criteria". Thus, this term includes both inclusion and exclusion criteria.

RESULTS

The results show the relevant information and are well structured.

DISCUSSION

Overall, this section has a good and logical sequence. However, I recommend the authors to include future lines of research at the end of the section.

CONCLUSIONS

The conclusion section should be clear and concise; it should respond to the proposed objective. The inclusion of irrelevant or misplaced additional information may reduce the visibility of the main conclusions and results. Therefore, phrases such as "Further economic evaluations derived from well-designed and better reported RCTs would allow health decision-makers to feel more confident in considering home healthcare interventions" make more sense in the Discussion section. Please consider this and amend this section accordingly.

Author Response

Point 1. INTRODUCTION - I congratulate the authors on this section. It is well written, and concise and includes all the information necessary to contextualize using up-to-date and relevant literature and justify the need for the review.

Response 1: We appreciate it.

Point 2. METHODS - The authors declare that they follow the PRISMA guidelines in this systematic review; however, any reference to these guidelines is included in the manuscript. Please insert the appropriate reference when referring to PRISMA.

Response 2: The reference regarding PRISMA is cited as suggested on the website: https://prisma-statement.org/PRISMAStatement/PRISMAStatement.

The reference is the 14 inserted on page 2, line 77 and now we have also added it on page 4, line 146.

“14. Page, M. J.;  McKenzie, J. E.;  Bossuyt, P. M.;  Boutron, I.;  Hoffmann, T. C.;  Mulrow, C. D.;  Shamseer, L.;  Tetzlaff, J. M.;  Akl, E. A.;  Brennan, S. E.;  Chou, R.;  Glanville, J.;  Grimshaw, J. M.;  Hrobjartsson, A.;  Lalu, M. M.;  Li, T.;  Loder, E. W.;  Mayo-Wilson, E.;  McDonald, S.;  McGuinness, L. A.;  Stewart, L. A.;  Thomas, J.;  Tricco, A. C.;  Welch, V. A.;  Whiting, P.; Moher, D. The PRISMA 2020 statement: an updated guideline for reporting systematic reviews. BMJ 2021, 372, n71.”

Point 3. Please rename the subsection "Criteria for considering studies included in this review" to "Eligibility criteria". Thus, this term includes both inclusion and exclusion criteria.

Response 3: It was done as suggested (Page 2, line 84).

Point 4. RESULTS - The results show the relevant information and are well structured.

Response 4: Thank you.

Point 5. DISCUSSION - Overall, this section has a good and logical sequence. However, I recommend the authors include future lines of research at the end of the section.

Response 5:  It was done as suggested (lines 359-365).

Point 6. CONCLUSIONS: The conclusion section should be clear and concise; it should respond to the proposed objective. The inclusion of irrelevant or misplaced additional information may reduce the visibility of the main conclusions and results. Therefore, phrases such as "Further economic evaluations derived from well-designed and better reported RCTs would allow health decision-makers to feel more confident in considering home healthcare interventions" make more sense in the Discussion section. Please consider this and amend this section accordingly.

Response 6: We created a new subtopic at the end of the discussion as suggested above (“4.3. Future research”, lines 359-365) and we moved this sentence from the conclusion to the end of the discussion.